# Interventions for the Prevention of Oral Mucositis in Patients Receiving Cancer Treatment: Evidence from Randomised Controlled Trials

Giuseppe Colella [1], Ciro Emiliano Boschetti [1], Rita Vitagliano [1], Chiara Colella [1], Lebei Jiao [2], Natalie King-Smith [2], Chong Li [2], Yii Nuoh Lau [2], Zacchaeus Lai [2], Ali Ibrahim Mohammed [2] and Nicola Cirillo [2,*]

1 Multidisciplinary Department of Medical-Surgical and Dental Specialties, Oral and Maxillofacial Surgery Unit, University of Campania "Luigi Vanvitelli", 80138 Naples, Italy
2 Melbourne Dental School, The University of Melbourne, Carlton, Melbourne, VIC 3053, Australia
* Correspondence: nicola.cirillo@unimelb.edu.au; Tel./Fax: +61-03-9341-1473

**Abstract:** Oral mucositis is a common and most debilitating complication associated with cancer therapy. Despite the significant clinical and economic impact of this condition, there is little to offer to patients with oral mucositis, and the medications used in its management are generally only palliative. Given that mucositis is ultimately a predictable and, therefore, potentially preventable condition, in this study we appraised the scientific literature to evaluate effective methods of prevention that have been tested in randomised controlled trials (RCTs). Published high-level evidence shows that multiple preventative methods are potentially effective in the prevention of oral mucositis induced by radiotherapy, chemotherapy, or both. Anti-inflammatory medications (including benzydamine), growth factors and cytokines (including palifermin), cryotherapy, laser-and-light therapy, herbal medicines and supplements, and mucoprotective agents (including oral pilocarpine) showed some degree of efficacy in preventing/reducing the severity of mucositis with most anticancer treatments. Allopurinol was potentially effective in the prevention of radiotherapy-induced oral mucositis; antimicrobial mouthwash and erythropoietin mouthwash were associated with a lower risk of development of severe oral mucositis induced by chemotherapy. The results of our review may assist in highlighting the efficacy and testing the effectiveness of low-cost, safe preventative measures for oral mucositis in cancer patients.

**Keywords:** oral mucositis; radiotherapy; chemotherapy; preventative agents

## 1. Introduction

Oral mucositis (OM) is a debilitating condition that affects patients undergoing cancer chemotherapy and/or radiotherapy to the head and neck region. OM is characterized by inflammation and ulceration of the oral mucosa, oral erythema, and pain. Of those undergoing chemotherapy, approximately 40% will go on to develop OM at some stage during treatment, typically 5 to 7 days after starting medication. For those undertaking both chemo- and radiotherapy, this percentage increases to approximately 90–100% [1,2]. Patients affected by OM may experience severe pain, difficulty in swallowing, taste changes, decreases in weight, and secondary infections [3]. These negative outcomes can interfere with cancer therapy and drastically impact the individual's quality of life, which is why adequate tools for the prevention of the condition are extremely important.

The pathogenesis of OM is complex, dynamic, and intricate. Both radiation- and chemotherapy-induced mucositis are initiated by basal epithelial cell death due to cellular damage by free radicals and inflammatory factors. This is followed by the upregulation of genes that increase the injury response by stimulating pro-inflammatory cytokines (e.g., interleukin 6), inflicting damage on the mucosal tissues. These signals are amplified, and as

the condition progresses, epithelial proliferation is reduced due to a loss of renewal capacity. Consequently, the oral epithelium begins to thin, and the early signs of the condition are experienced by patients, such as ulcerations and erythema [2–4].

OM can be assessed using a variety of grading systems; however, it is commonly classified according to the World Health Organisation (WHO) grading system. This system scores OM presentation and symptoms based on severity, from grade 0 to IV. Grade 0 indicates no objective findings of OM. Grade I implies the presence of painless ulcerations, erythema, or mild discomfort with no lesions. Grade II denotes the occurrence of painful ulcerations or erythema that do not impose on the patient's eating abilities. Grade III indicates painful confluent pseudomembranous ulcerations or erythema that impact the patient's ability to eat and drink. Grade IV implies severe deep and/or necrotic ulcerations that require enteral or parenteral support [5,6].

In the last two decades, many treatment techniques for radio–chemotherapy-induced oral mucositis in cancer patients have been established. After intensive reviews of clinical trials involving different modalities for the management of mucositis, various international organizations have published their recommendations for managing oral mucositis. These include the Mucositis Study Group of the Multinational Association of Supportive Care in Cancer/International Society of Oral Oncology (MASCC/ISOO), National Comprehensive Cancer Network (NCCN), European Society of Medical Oncology (ESMO), European Oncology Nursing Society (EONS), and Cochrane reviews. Despite these considerable efforts, there is little to offer to patients with oral mucositis, and the medications used in its management remain generally only palliative.

Given that the nature of OM is fundamentally iatrogenic, it is reasonable that attempts have been made to prevent this serious complication of cancer treatment. Presently, a considerable body of scientific literature is available that describes possible interventions for preventing OM; however, due to the lack of solid evidence, the vast majority may not be effective or appropriate for a specific anticancer regimen. For this reason, we undertook a comprehensive assessment of the literature that included only the highest level of evidence, namely, randomised controlled trials (RCTs). A structured search strategy conducted on the PubMed and Scopus databases was developed by combining the search terms related to three categories, namely: 1. Mucositis ("mucositis" [Title/Abstract] OR "mucosal injury" [Title/Abstract]); 2. Oral cavity ("oral"[Title/Abstract] OR "mouth" [Title/Abstract] OR "oral cavity" [Title/Abstract]); Cancer treatment ("chemotherapy" [Title/Abstract] OR "radiotherapy" [Title/Abstract] OR "cancer treatment" [Title/Abstract]). The articles screened from this search (in the English language only) were used as the basis for the elaboration of the manuscript. Further grey literature was searched as appropriate.

Our review drawing from the results of RCTs shows that while several agents are potentially effective in reducing the incidence and severity of oral mucositis, there is considerable heterogeneity in the published data.

## 2. Overview of Available Preventative Strategies for Oral Mucositis

The preventative strategies that were assessed in this review included anti-inflammatory and anti-microbial medications, growth factors and cytokines, laser and light therapy, cryotherapy, herbal medications, supplements, muco-protective agents, and others (Table 1). These were then stratified into radiotherapy, chemotherapy, or chemoradiotherapy to identify whether certain treatments may be better suited depending on the type of cancer therapy being undertaken by the patient. Most evidence collected was based on the results of RCTs for chemotherapy-induced OM, whereas fewer studies examined preventative strategies for radiotherapy-induced OM. Detailed description of each individual study included in this review can be found in Tables S1–S3.

Reduction in severity and/or duration of OM were recorded for a number of medications, including: anti-inflammatory benzydamine, laser-and-light therapy, allopurinol, cryotherapy, multiple herbal agents, supplements, and mucoprotective agents (radiotherapy-induced OM; Table S1); antimicrobial mouthwash, erythropoietin mouthwash, low-level laser

therapy, cryotherapy, multiple herbal agents, supplements, pilocarpine, and other agents (chemotherapy-induced OM; Table S2); anti-inflammatory medications, laser-and-light therapy, herbal medicines, supplementation, mucoprotective agents, and other agents (chemoradiotherapy-induced OM; Table S3).

**Table 1.** Sample of studies with significant reduction in severity or incidence of oral mucositis. OM, oral mucositis; RT, radiotherapy; CT, chemotherapy; CRT, chemoradiotherapy.

| Treatments | Interventions | Significant Reduction in Severity of OM (Ref. n.) | Significant Reduction in Incidence of OM (Ref. n.) |
|---|---|---|---|
| RT | Anti-inflammatory medication | [7–9] | [9] |
| | Laser and light therapy | [10–13] | [12,13] |
| | Herbal | [14–17] | [18,19] |
| | Supplement | [20,21] | N/A |
| | Mucoprotective | [22,23] | N/A |
| | Others | [24–26] | [25] |
| CT | Antimicrobial medication | [27–30] | [28,29] |
| | Growth factors and Cytokines | [31,32] | [31,32] |
| | Laser and light therapy | [33–36] | [33,36] |
| | Cryotherapy | [37] | [37] |
| | Herbal | [38–40] | [41–43] |
| | Supplement | [44–52] | [44–47,53–55] |
| | Mucoprotective | [56,57] | [57] |
| | Others | [58–60] | 81 |
| CRT | Antimicrobial medication | [61] | N/A |
| | Growth factors and Cytokines | [62,63] | [62,63] |
| | Laser and light therapy | [36,64–67] | [36,64,65] |
| | Herbal | [68,69] | [70] |
| | Supplement | [71–73] | [71–73] |
| | Mucoprotective | [74] | N/A |
| | Others | [75,76] | N/A |

Importantly, our analysis identified a significant level of heterogeneity in the published data, making it difficult to compare and evaluate the quality of evidence. Different assessment protocols were utilized to test the efficacy of the preventative intervention in the studies: these were based on different parameters (e.g., pain, weight loss, severity), different scoring system and scale (e.g., OMAS, CTCAE, and WHO) and different dosages of the same drugs. For many of the novel interventions, there are only one or two articles supporting their effects; therefore, the efficacy of these interventions is debatable and further investigation is warranted. Due to the severity of OM and cancer itself, many subjects (up to 50%) were lost to follow up in some of the studies, hence reducing the validity of those studies. This heterogeneity reduces the generalizability of the evidence presented in this review.

*2.1. Anti-Inflammatory Medications*

Inflammation has significant implications in radiotherapy and chemotherapy-induced OM, with pro-inflammatory cytokines thought to be a contributing factor to the pathogenesis of the condition. Therefore, the use of anti-inflammatory agents was proposed to prevent and alleviate OM [77]. In the anti-inflammatory medications group, only benzy-

damine was examined in RCTs. This intervention was only used in those being treated with radiotherapy or chemoradiotherapy and not chemotherapy alone. However, the results of all studies showed statistically significant results, with benzydamine being effective in the prevention of radiotherapy and chemoradiotherapy induced OM [7–9,61].

### 2.2. Antimicrobials

The current research suggests that microbial colonisation is an important factor in the propagation of the inflammatory response seen in OM. Hence, antimicrobials were proposed as an effective method to prevent the clinical manifestations of OM [78]. In the antimicrobial medications group, chlorhexidine was the main drug studied for those undergoing chemotherapy and chemoradiotherapy. The results were varying, with three studies [79–81] reporting no statistically significant differences between the groups, and three studies [27,28,58] suggesting that there were statistically significant differences favouring the chlorhexidine group. The only study investigating patients undergoing chemoradiotherapy was Diaz-Sanchez et al. [81], with the rest only examining chemotherapy patients. Three studies [30,82,83] investigated different antimicrobial medications, which were Iseganan, azithromycin, and povidone-iodine, respectively. Statistically significant results were only found in the azithromycin group. Hence, the data supporting the use of antimicrobials for the prevention of OM are contrasting or inconclusive.

### 2.3. Herbal Medications

There is a growing interest in the therapeutic benefits of herbal medicine on cancer-therapy-induced OM due to their reduced side effects, cost, and better availability compared to synthetic drugs [84]. Herbal agents exert their effects via several mechanisms of action consisting of antioxidant, analgesic, anti-inflammatory, antifungal, antiseptic, and anti-carcinogenic activity [85]. Three studies found that curcumin possesses anticancer and anti-inflammatory effects that make it an effective agent in preventing and alleviating OM induced by radiotherapy and chemoradiotherapy [14,69,70]. The results showed significant biological effects of curcumin on prevention and treatment of OM. Two studies looked at the effect of oral silymarin on prevention of radiotherapy-induced OM: one [15] reported significant prophylactic effects as well as alleviation of OM by silymarin oral tablets while the other one [86] was unable to provide evidence that a nano-solution of silymarin prevented OM in patients receiving radiotherapy. Other studies on prevention of radiotherapy-induced OM evaluated the therapeutic potential of Black Mulberry Molasses; Glycerin payayor, a Persian medicine herbal compound; Qingre Liyan decoction; and honey–lemon spray [16–19,87]. Significant effects were found for all of the herbal agents mentioned above except honey–lemon spray.

Chamomile, characterized by its anti-inflammatory effects, has a long history of use in treatment of inflammatory conditions of the skin and mucosa [88]. The results from two studies of chamomile in the form of cryotherapy and topical gel showed a significant reduction in the occurrence and severity of chemotherapy-induced OM [39,41]. Conversely, one study [89] failed to demonstrate the benefit of chamomile mouthwash. Upon an exploratory subset analysis based on gender, the authors revealed that chamomile might be beneficial for males and detrimental for females. Further research is needed to confirm this clinically important hypothesis. The use of other herbal agents such as aloe vera, honey, Persica, Plantago ovata hydrocolloid, and Quercetin for patients undergoing chemotherapy were assessed by multiple studies [38,40,42,43,90]. The results identified a significant effect of a topical aloe vera solution, honey and Plantago ovata hydrocolloid in the prevention and reduction of chemotherapy-induced OM [38,40,42]. Conversely, other studies concluded that there is a lack of evidence for the efficacy of Persica oral drops and Quercetin in chemotherapy patients [43,90].

One study [68] that focused on the prevention of chemoradiotherapy-induced OM concluded that calendula officinalis mouthwash is effective in reducing OM severity but is unable to fully prevent its occurrence.

### 2.4. Laser Therapy

Low level laser therapy (LLLT) is a non-invasive management tool that involves the local application of low-level lasers or light-emitting diodes to the oral mucosa [91]. The use of LLLT induces an analgesic, anti-inflammatory, and wound-healing effect in the tissue via the modulation of reactive oxygen species and activation of transcription factors, leading to increased cell proliferation and migration [92]. The results from studies are highly consistent in recognising the beneficial effect of LLLT as a preventative strategy for OM induced by chemotherapy [33–36,93,94], radiotherapy [10–13], or radio-chemotherapy [36,64–67]. Only one study [95] found no benefit from prophylactic use of LLLT in patients (3 to 18 years-old) with cancer treated with chemotherapy. The conflicting results in this paper might be attributed to the younger subject population, namely, children and adolescents. Overall, these results are consistent with the guidelines by the Multinational Association of Supportive Care in Cancer and International Society of Oral Oncology (MASCC/ISOO) Clinical Practice Guidelines for Mucositis 2021, which recommended a preventive low-level laser therapy in patients undergoing high-dose chemotherapy, radiotherapy, or radiotherapy with chemotherapy [96]. Therefore, evidence demonstrates that LLLT can potentially be used as a routine preventative intervention for OM in patients undergoing chemotherapy, radiotherapy, or combination therapy.

### 2.5. Cryotherapy

Cryotherapy frequently describes cooling of the mouth during chemotherapy to prevent OM. The common theory for its mechanism is that it promotes vasoconstriction, leading to reduced local drug concentration and toxicity during chemotherapy infusion [97,98]. The current literature focuses mainly on oral cryotherapy for the prevention of chemotherapy-induced mucositis, as compared to radiotherapy. Conflicting results were reported in the three studies included [37,99,100]. Owing to the limited evidence, no conclusion can be made regarding the efficacy of oral cryotherapy as a prophylactic tool for OM during chemotherapy. However, given that cryotherapy using ice is well tolerated and inexpensive, further studies are warranted, particularly in patients receiving radiotherapy in the head and neck region.

### 2.6. Growth Factors

Palifermin (keratinocyte growth factor, KGF) is the prototype of an anti-mucositis drug and was first shown to reduce the duration and severity of oral mucositis after intensive chemotherapy and radiotherapy for hematologic cancers [101]. The efficacy of palifermin in preventing OM was subsequently confirmed in patients with solid tumours receiving either chemotherapy [62] or chemoradiotherapy [32]. In one study, single-dose palifermin was found to prevent severe oral mucositis during doxorubicin-based chemotherapy in patients with sarcoma [63]. However, palifermin failed to reduce OM in a chemotherapy-only, high-dose melphalan transplant setting, suggesting that its efficacy is dependent upon the type of cancer treatment [21]. Furthermore, it is not uncommon to find that physician-assessed mucositis is not paralleled by a better patient-reported outcome, assessed through the mouth and throat soreness score of patients under palifermin vs. placebo arms [32]. It also remains to be seen whether palifermin promotes tumour growth in epithelial cancers. Currently, palifermin is approved for use by FDA only in patients with cancer who receive high doses of chemotherapy and radiation therapy followed by stem cell rescue.

Granulocyte-macrophage colony-stimulating factor (GM-CSF) is a glycoprotein produced by human cells that stimulates the proliferation and maturation of cells including granulocytes, macrophages, and eosinophils [102]. The studies included in this review demonstrated that GM-CSF was not effective for the prevention of radiotherapy-induced OM [103,104] or chemotherapy-induced OM [105]. Furthermore, there is evidence suggesting that GM-CSF may be associated with reduced local tumour control in patients undergoing head and neck radiotherapy [106]. Hence, the use of GM-CSF as a prevention of OM as a treatment is questionable.

Owing to the possible side-effects of systemic administration, topical interventions have also been developed. Transforming growth factor (TGF)-beta3 used as mouthwash (25 microg/mL of TGF-beta3, 10-mL per application) was safe but ineffective compared to placebo prevention or alleviation of chemotherapy-induced oral mucositis in patients with lymphomas or solid tumours [107]. Conversely, patients undergoing autologous hematopoietic SCT who received erythropoietin (EPO) mouthwash (50 IU/mL, 15 mL four times a day) from the starting day of high-dose chemotherapy had reduced incidence and duration of OM [31]. However, the evidence from the RCTs available with regards to the use of TGF-β3 and EPO mouthwash is not sufficient to draw a conclusion about their efficacy in the prevention of OM.

### 2.7. Supplements

Of the studies investigating supplement interventions, five studies examined zinc sulphate and six studies examined glutamine; the remaining studies that examined novel supplement interventions have either inconclusive/conflicting evidence [20,48,108–111] or are potentially promising interventions [44,45,53,55,73,112]. However, as these interventions were only examined by one or two studies, further investigation is needed to determine their effectiveness.

Studies have mentioned that zinc supplementation favours ulcer healing, has an anti-inflammatory effect, and assists with mucosal recovery [50]. Four of the five studies [72–75] identified a significant decrease in mucositis intensity and severity. However, Mansouri et al. [113] were unable to identify any significant decrease with zinc supplementation, mentioning that the concentration of the intervention might be too low to demonstrate a significant difference. Additionally, due to the acute damaging effect of CT, an excessive amount of zinc is released from the tissues and zinc supplementation might not be enough to produce an immediate healing effect due to the significantly decreased zinc concentration in the tissue [113].

Glutamine has a protective effect against mucosal damage by reducing proinflammatory cytokine production and its metabolite glutathione regulates intracellular redox potential, thus, buffering the action of reactive oxygen species that play a critical role in the initiation of OM [67]. Of the six articles [44,46,47,71,72,114] that investigated the effects of glutamine in OM prevention, four [46,47,71,72] identified a significant decrease in the frequency of OM occurrence as well as a delay in OM development. The two studies [44,114] that did not identify a significant difference in the occurrence and severity of OM mentioned that the results may have been limited to the amount of glutamine given to achieve significance. Furthermore, it is to be noted that OM occurrence and severity were lower than controls.

### 2.8. Mucoprotective Agents

Sucralfate has been effectively used to treat ulcers in the gastrointestinal tract and, therefore, was suggested as a potential preventative agent for both chemo/radiotherapy-induced OM. However, all articles have suggested that sucralfate only reduces the severity/grade of OM or pain and there is no evidence supporting the reduction of the incidence rate of radiotherapy-induced OM. Therefore, it is safe to say that sucralfate may only be an effective means to improve the function and symptoms, but there is no evidence supporting the preventative effectiveness of sucralfate regarding OM caused by chemo/radiotherapy.

Studies may also present with false-negative results as many were underpowered and/or used inadequate measuring and assessment tools, as reported by Nottage et al. [115]. There is also a lack of clear instruction concerning the specific concentration of sucralfate that was administered to the intervention group, only the volume. Furthermore, studies conducted with small sample sizes may contribute to sampling errors [23].

Oral pilocarpine (OP) and a rebamipide gargle were also investigated. Pilocarpine stimulates the secretion of saliva, which works as a natural mucoprotective agent. The study showed OP was able to decrease the incidence of OM and severe OM very significantly

in patients who underwent chemotherapy, thus suggesting OP as an extremely potent prophylactic medication. The finding is consistent with the results reported by other articles that suggested OP may be effective for prevention and treatment of radio/chemotherapy-induced xerostomia [116]. Further research should focus on the preventative effectiveness of OP for patients who suffer radiotherapy-induced OM.

The study on the rebamipide gargle, however, did not show any clinical preventative effect against chemoradiotherapy-induced OM [74].

*2.9. Other Medications*

Low-temperature atomization inhalation (LTAI) has demonstrated a preventative effect against radiotherapy-induced OM [25]. It is thought that low temperature inhibits the inflammatory response, reduces mucosal oedema, and decreases pain. This is consistent with an in vitro study where results reported increased cell viability and reduced cytokine production (IL-6 and TNF-$\alpha$) at low temperatures [117].

Zinc chloride used as a mouthwash determined a reduction in the incidence of chemotherapy-induced OM [59]; therefore, it is believed that zinc, with its ability to increase protein synthesis and improve cell membrane stability, may be potentially used as a prophylactic medication for chemotherapy-induced OM. However, one obvious limitation with this study is that there is a small sample size. Future studies on zinc chloride mouthwash should aim to increase the sample size to achieve adequate power and statistical significance. The application of recombinant human intestinal trefoil factor (rhITF) [58] and education therapy have only improved the severity and quality of life for patients; therefore, they should only be applied as treatment rather than prevention.

Intravenous Actovegin [75] and a comprehensive oral care regimen only exhibited effects in reducing the incidence of severe chemoradiotherapy-induced OM, whereas pentoxifylline [118] was ineffective with no preventative effect observed. Hence, there is a lack of evidence supporting their usage as prophylaxis for OM.

Recently, melatonin (20 mg) used in addition to conventional treatment (anti-fungal and anti-inflammatory agents, topical anaesthetics) has shown controversial results [119].

**3. Limitations**

This study does not come without limitations. The use of two search engines limited to the English language for peer-reviewed articles and the selection of a specific search string may have failed to identify relevant studies. We acknowledge that there may also be limitations to the internal and external validity of the present report. Despite using a reproducible search, this was fundamentally a narrative review where the opinions of experts may have come to light. Furthermore, we did not assess the risk of bias of the studies included. Nevertheless we believe that this review provided useful insights in that it scrutinized the highest level of evidence from primary studies assessing preventative interventions for oral mucositis in cancer patients.

It is important to point out that the results of individual studies with specific cancer regimens and schedules may not be applicable to others. For example, palifermin failed to reduce OM in a high-dose melphalan transplant setting, suggesting that its efficacy is dependent upon the type of cancer treatment [21]. A recent network meta-analysis demonstrated palifermin may be inferior to cryotherapy in reducing the risk of chemotherapy-induced oral mucositis and is associated with a considerable higher risk of taste disturbance [120].

Another limit to the external validity of the studies reported in this review is the target population. For example, most RCTs were undertaken in adults, which limits the transferability of the findings to children undergoing cancer treatment. For example, a recent clinical practice guideline [121] recommended cryotherapy for older cooperative paediatric patients who will receive short infusions of melphalan or 5-fluorouracil. The same authors conclude that intraoral photobiomodulation therapy (620–750 nm spectrum) should be used in paediatric patients undergoing autologous or allogeneic HSCT and for paediatric

head and neck carcinoma patients undergoing radiotherapy. This guideline indicates that palifermin should not be used routinely in paediatric cancer or HSCT patients [121].

## 4. Conclusions and Future Directions

Mucositis remains a significant burden for patients who are undergoing anti-neoplastic drugs or radiation therapy [122]. Here, we highlighted that a number of agents are available and are potentially effective in the prevention of OM associated with cancer treatment. Benzydamine was shown as an effective anti-inflammatory for the prevention of radio- and chemoradiotherapy induced OM. The results for antimicrobials and growth factors were varying and sometimes inconclusive. Low level laser therapy was found to be effective in preventing OM induced by all cancer treatments. Most herbal medicines and supplement interventions included in this review showed some level of efficacy in the prevention of OM. Mucoprotective agents remain largely ineffective as a prevention for chemo/radiotherapy, with the exception of oral pilocarpine. Zinc chloride has also demonstrated a prophylactic effect. However, these interventions were only assessed by a small number of studies identified by this review and hence further investigation is warranted to test their true efficacy. Notwithstanding the limitations, the results of our comprehensive investigation may assist in paving the way for future management options, which is vital considering the adverse consequences associated with the condition.

A better understanding of the pathophysiology of mucositis is instrumental for the development and testing of novel mechanism-based drugs for this condition [3]. It will be important to develop intervention that reduce both oral and intestinal mucositis, which are part of the same disease spectrum. Considering that at the molecular level, chemo/radiotherapy-induced mucosal injury is associated with the production of reactive oxygen species (ROS) at early preclinical stages [123], inhibition of this pathway could be a valid prophylactic and/or therapeutic strategy.

**Supplementary Materials:** The following supporting information can be downloaded at: https://www.mdpi.com/article/10.3390/curroncol30010074/s1, Table S1: Summary of articles reporting preventative interventions for radiotherapy-induced OM. Table S2: Summary of articles reporting preventative interventions for chemotherapy-induced OM. Table S3: Summary of articles reporting preventative interventions for chemoradiotherapy-induced OM.

**Author Contributions:** Conceptualization and methodology, N.C. and G.C.; software, C.C.; validation, C.E.B., C.C. and R.V.; investigation and formal analysis, L.J., N.K.-S., C.L., Y.N.L. and Z.L.; data curation, C.E.B. and R.V.; writing—original draft preparation, L.J., N.K.-S., C.L., Y.N.L., Z.L., A.I.M. and N.C.; writing—review and editing, R.V. and G.C.; supervision, N.C. and G.C.; project administration, N.C. All authors have read and agreed to the published version of the manuscript.

**Funding:** This research received no external funding.

**Acknowledgments:** The authors would like to acknowledge the support of The University of Melbourne and STEMM Research (UK) for making the necessary resources available.

**Conflicts of Interest:** The authors declare no conflict of interest.

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
