# Peer review of "Interventions for the Prevention of Oral Mucositis in Patients Receiving Cancer Treatment: Evidence from Randomised Controlled Trials"

_curroncol, doi:10.3390/curroncol30010074_

Round 1

Reviewer 1 Report

Dear authors,

many compliments for your work that, even if is a narrative review about the topic, it should be pointed out that it has been conducted following a reproducible and well-organized search strategy so, in my opinion, it can be considered as a strong step forward for the topic.

  The manuscript is review on RCTs about the prevention of oral mucositis in patients treated for cancer. Oral mucositis represents one of the most usual adverse effects of cancer treatments and the definition of an effective strategy for the prevention and treatment of such effects is very relevant in the field. There is a general lack of evidence about the topic so this manuscript, in the form of a review summarizing the existing literature, provides an interesting and reliable tool for enlightening the topic. The methodology is excellent, no particular suggestions can be given. Conclusions, references and tables and figures are appropriate.

Regards

Author Response

Many thanks for your positive comments

Reviewer 2 Report

Comments for the Authors   GENERAL

This review attempts to evaluate 16 effective methods of prevention of oral mucositis that have been tested in randomised controlled trials (RCTs).

 Here are my observations on this study/paper:

Abstract

Entirely appropriate.
Aims and objectives of the review are not clear

Introduction

·       Lack essential details.

·       There are a number of up to date references which should be included and omit the entire outdated one.

·        Aims and objective of the study are not clearly stated.

Material and Methods

·       Not exhaustive as it was limited to only two databases.

Results

Appropriate

Discussion

  • The present study has several limitations. First, the lack of in-depth details and the findings are limited to data generated from only two databases, this may ignore important and relevant publications in the field published elsewhere and thus jeopardize the conclusion reached.
  • All these limitations should be highlighted at the start of the discussion.
  • It may be helpful for the reader if you summarize the findings in tables and provide brief discussion.

 Conclusion

·       The conclusions are questionable in light of the above-mentioned limitations.

References

·       Some of the references do not conform to the journal style.

Author Response

Comments attached
